# Integration of Unmanned Aerial Vehicle Imagery and Machine Learning Technology to Map the Distribution of Conifer and Broadleaf Canopy Cover in Uneven-Aged Mixed Forests

**Nyo Me Htun [1], Toshiaki Owari [2],\*, Satoshi Tsuyuki [1] and Takuya Hiroshima [1]**

[1] Department of Global Agricultural Sciences, Graduate School of Agricultural and Life Sciences, The University of Tokyo, Tokyo 113-8657, Japan; nyomehtun209@g.ecc.u-tokyo.ac.jp (N.M.H.); tsuyuki@fr.a.u-tokyo.ac.jp (S.T.); hiroshim@g.ecc.u-tokyo.ac.jp (T.H.)

[2] The University of Tokyo Hokkaido Forest, Graduate School of Agricultural and Life Sciences, The University of Tokyo, Furano 079-1563, Hokkaido, Japan

\* Correspondence: owari@g.ecc.u-tokyo.ac.jp

**Abstract:** Uneven-aged mixed forests have been recognized as important contributors to biodiversity conservation, ecological stability, carbon sequestration, the provisioning of ecosystem services, and sustainable timber production. Recently, numerous studies have demonstrated the applicability of integrating remote sensing datasets with machine learning for forest management purposes, such as forest type classification and the identification of individual trees. However, studies focusing on the integration of unmanned aerial vehicle (UAV) datasets with machine learning for mapping of tree species groups in uneven-aged mixed forests remain limited. Thus, this study explored the feasibility of integrating UAV imagery with semantic segmentation-based machine learning classification algorithms to describe conifer and broadleaf species canopies in uneven-aged mixed forests. The study was conducted in two sub-compartments of the University of Tokyo Hokkaido Forest in northern Japan. We analyzed UAV images using the semantic-segmentation based U-Net and random forest (RF) classification models. The results indicate that the integration of UAV imagery with the U-Net model generated reliable conifer and broadleaf canopy cover classification maps in both sub-compartments, while the RF model often failed to distinguish conifer crowns. Moreover, our findings demonstrate the potential of this method to detect dominant tree species groups in uneven-aged mixed forests.

**Keywords:** species group classification; uneven-aged mixed forest; UAV imagery; U-Net algorithm; random forest

## 1. Introduction

The classification and mapping of vegetation communities is an essential task for the successful management of forests worldwide [1–3]. In particular, uneven-aged mixed forests have unique characteristics such as species diversity, stand heterogeneity, and irregular distribution of trees, leading to their recognition as important contributors to biodiversity conservation, ecological stability, carbon sequestration, the provisioning of ecosystem services, and sustainable timber production. Thus, it is essential to elucidate forest dynamics and develop quantitative methods for analyzing the impacts of factors such as climate change on such forests [4]. For this reason, societies have used forest maps to understand and manage the distribution of vegetation communities for hundreds of years [5,6]. When developing forest maps, representing the current forest condition is essential to initiate appropriate protection and restoration programs. However, preparing a correct forest map using traditional field surveys is time-consuming and costly [7].

Remote sensing techniques offer an alternative method to obtain highly accurate forest composition information [8]. As remote sensing technology can efficiently cover

large-forested areas, providing a large amount of forest information in a relatively short amount of time, it is an efficient and effective method for forest mapping [1]. Some studies have demonstrated the application of high- or very-high-resolution remote sensing data (e.g., WorldView-2, RapidEye satellite imagery data) for mapping of forest-covered areas. However, the high cost of data acquisition of such methods is prohibitive for the monitoring of large-scale forest areas [9]. For broad forest cover classification, some open-source satellite imagery, such as Sentinel-2 products, can be used with reliable accuracy. However, for deeper classification processes such as typical tree species group classification and individual tree species identification, very-high-resolution remote sensing datasets are a prerequisite. Over the last decade, many researchers and professionals have increasingly recognized the applicability of unmanned aerial vehicles (UAVs), also known as drones, to mapping of vegetation species and communities [10–13]. Specifically, UAVs have gained popularity for vegetation mapping due to their versatility, cost-effectiveness, and ability to capture high-resolution imagery and data in a noninvasive manner [14–16].

Meanwhile, recent studies have demonstrated the applicability of machine learning technology to classification problems in many sectors, including the forestry sector [17] due to its adaptability, scalability, and capacity to learn patterns and relationships from data and then make predictions or classifications based on that learning. Many studies have described the application of semantic segmentation-based machine learning models to various forest planning operations such as forest cover mapping, forest type and tree species classification, and forest fire detection [17–19]. Semantic segmentation involves assigning a categorical label to every pixel in an image. Specifically, this method is based upon the usual image segmentation method, wherein a semantic description is added to the target and background image. The output of a semantic segmentation model is a pixel-level label map in which a class label is assigned to every pixel of the image.

To date, studies focused on the integration of UAV data with semantic segmentation-based machine learning algorithms for mapping of uneven-aged mixed forests remain limited. In particular, the performance of these integrated technologies at identifying tree species groups in uneven-aged mixed forests has rarely been studied.

Therefore, this study was conducted to investigate the feasibility of integrating UAV imagery and semantic segmentation-based machine learning classification algorithms to describe conifer and broadleaf canopy distribution in uneven-aged mixed forests of the University of Tokyo Hokkaido Forest. To achieve this aim, we selected the most reliable model from two machine classification algorithms (U-Net and random forest) for integration with the UAV dataset. Moreover, we investigated the importance of data balancing (balanced pixel numbers between conifer and broadleaf classes) to improve the accuracy of machine learning models trained on the UAV dataset.

## 2. Materials and Methods

### 2.1. Study Site

This study was conducted at the University of Tokyo Hokkaido Forest (UTHF) (43°10–20′ N, 142°18–40′ E, 190–1459 m asl), located in Furano City in the central part of Hokkaido Island, northern Japan (Figure 1a,b). The UTHF is an uneven-aged mixed coniferous and broadleaf forest located in the transition zone between cool temperate deciduous forests and hemiboreal coniferous forests. The total area of UTHF is 22,717 ha and the predominant species include *Abies sachalinensis*, *Picea jezoensis*, *Acer pictum* var. *mono*, *P. glehnii*, *Fraxinus mandshurica*, *Kalopanax septemlobus*, *Quercus crispula*, *Betula maximowicziana*, *Taxus cuspidata*, and *Tilia japonica*. Moreover, the dwarf bamboo (*Sasa senanensis* and *S. kurilensis*) is frequently present on the forest floor of UTHF. The mean annual temperature and annual precipitation at the arboretum were 6.4 °C and 1297 mm (230 m asl), respectively, in 2001–2008. The ground is usually covered with snow from late November to early April to a depth of about 1 m [20,21].

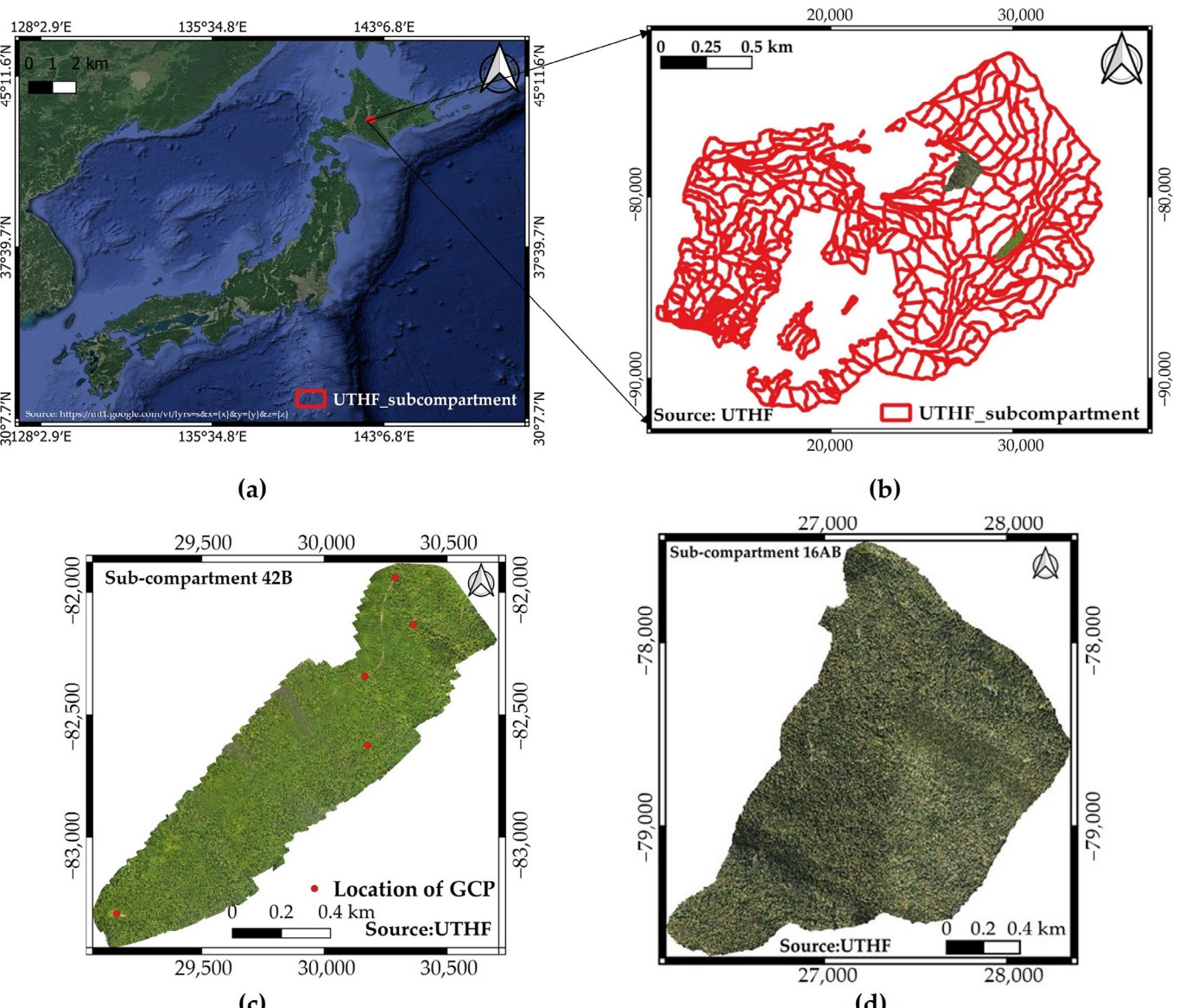

**Figure 1.** (**a**) Location map of the study area in Japan; (**b**) location map of the selected two Sub-compartments in the University of Tokyo Hokkaido Forest; aerial orthomosaics of Sub-compartment (**c**) 42B and (**d**) 16AB.

Specifically, the present study was conducted across two sub-compartments of the UTHF: Sub-compartment 42B with an area of 90.3 ha, which is mainly occupied by broadleaf tree species (Figure 1c), and Sub-compartment 16AB with an area of 150 ha, where conifer tree species are dominant (Figure 1d).

### 2.2. Acquisition of UAV Imagery

Aerial images were collected separately in the two sub-compartments, with different UAV platforms and image resolutions. The flight operation was conducted at Sub-compartment 42B on 5 August 2022 using the DJI-Inspire 2 UAV platform equipped with a red–green–blue (RGB) camera (Figure 2a), while operations at Sub-compartment 16AB on 28 October 2022 employed the DJI Matrice 300 RTK UAV platform equipped with a DJI Zenmuse P1 RGB camera with 35-megapixel sensor (Figure 2b). The weight of the former UAV is 3.4 kg including the battery but excluding the camera and that of the latter is 6.3 kg (including two TB60 batteries).

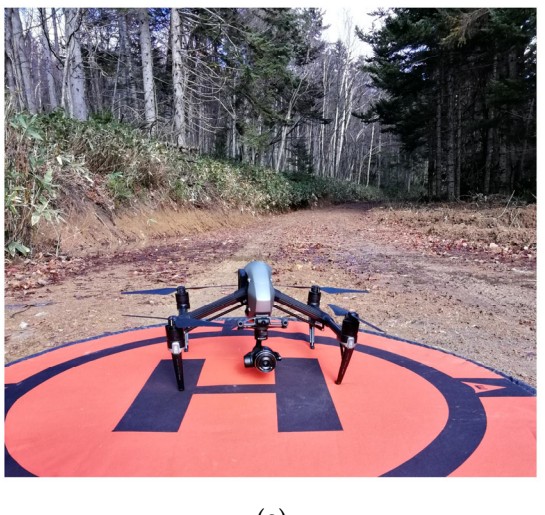
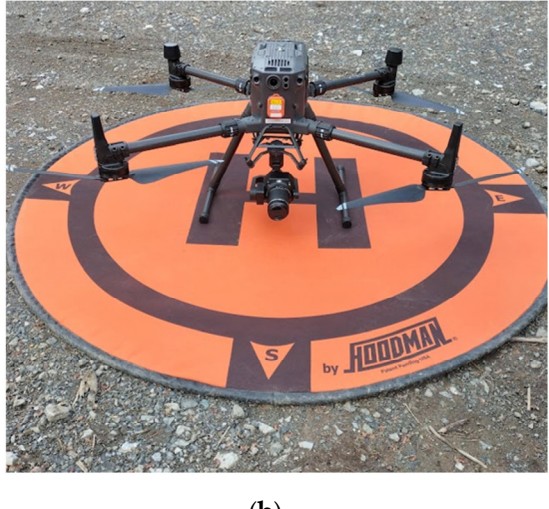

(**a**)                          (**b**)

**Figure 2.** (**a**) DJI-Inspire 2 UAV platform used in Sub-compartment 42B; (**b**) DJI Matrice 300 RTK UAV platform used in Sub-compartment 16AB.

The UAV flights were programmed to follow a predetermined square parallel flight plan covering the study sites by flying at a fixed altitude of approximately 130 m (above ground level) at Sub-compartment 42B with a flying speed around 7 m/s and at 200 m (above ground level) at Sub-compartment 16AB with a maximum flying speed of 26 m/s. Although according to the Flight Rules for a UAS adopted by Ministry of Land, Infrastructure, Transport, and Tourism (MLIT), Japan, the flight altitude for a UAS in Japan must not be higher than 150 m above ground level, special permission can be applied for research or scientific studies (MLIT, https://www.mlit.go.jp/en/koku/uas.html, accessed on 30 November 2023). In our case, due to the undulating topography of our study sites and the limited time available to fly the UAV over the entire study area, we applied for this special permit. The aerial images were acquired in both sub-compartments at 3 s intervals to achieve overlap along the route of 85% and 80% and overlap between routes of 82% and 90% for Sub-compartment 42B and Sub-compartment 16AB, respectively. The image and ground resolution of the Sub-compartment 42B imagery were 5280 × 3956 (pixels) and 2.5 cm/pixel, while those of Sub-compartment 16AB were 8192 × 5460 (pixels) and 1.6 cm/pixel. Prior to the UAV flight in Sub-compartment 42B, five ground-control points (GCPs) were set up across the study sites, particularly at locations that are easily visible from the sky (Figure 1c). The GCP locations were recorded with a real-time kinematic (RTK) dual-frequency global navigation satellite system receiver (DG-PRO1RWS, BizStation Corp., Matsumoto City, Japan). In the case of Sub-compartment 16AB, we did not establish GCPs because the dual-frequency RTK GNSS in the Matrice 300 allows direct georeferencing of UAV imagery without the need for GCPs [22].

### 2.3. Data Analysis

#### 2.3.1. Processing of UAV Imagery

The collected aerial images were processed using Pix4Dmapper version 4.8.0 (Lausanne, Switzerland). The software calculates the camera positions and orientations in three-dimensional space through analysis of feature points on different images through the process known as 'structure from motion' to generate orthomosaic images of the study areas. This workflow involved importing the UAV images, alignment of the images and calibration, mesh building, construction of the dense point cloud, and generation of the orthomosaic. In the case of Sub-compartment 42B, GCPs were also used to improve the accuracy of photogrammetric products at the image alignment stage (GCP coordinates were imported into Pix4Dmapper and then each GCP was identified manually).

The orthomosaics of the study areas thus generated were used for the following classification tasks. We used orthophotos of two sub-compartments with different resolutions: 11 cm/pixel for Sub-compartment 42B and 2.4 cm/pixel for Sub-compartment 16AB, which we resized into 7 cm/pixel to save computer memory.

### 2.3.2. Labelling of Images

Labkit, a Fiji plugin, was used to label the orthophotos of the study areas. Labkit is designed to handle large image data using consumer-level laptops or workstations. It brings together advanced image processing capabilities (ImgLib2), efficient large-scale image visualization (BigDataViewer), and incorporates a novel implementation of random forest pixel classification. Initially users need to draw a few pixel-based labels per class manually over the image and compute a feature vector for each labelled pixel through a configurable set of image filters such as such as Gaussian, difference of Gaussians or Laplacian filter. Then, the random forest classifier is trained on the feature vectors of labeled pixels. Hence, the classifier predicts the feature vectors of all pixels in the whole image and thereby generates the dense labelled mask (corresponding mask) [23].

In this study, we cropped orthophotos of the study areas into small patches of $512 \times 512$ pixels to avoid overloading the computer memory; this process created 360 images of Sub-compartment 42B and 1600 images of Sub-compartment 16AB.

After segmenting all images of both sub-compartments using Labkit, the generated masks were transferred into label layers. The labels include conifer class (coniferous species group), broadleaf class (broadleaf species group), and background, which includes all pixels in an image that are not part of a detected object or region of interest (i.e., areas between and around conifer and broadleaf canopies as well as the background area of the image in this study). They were denoted as class 2, 1 and 0, respectively.

### 2.3.3. U-Net Algorithm

The U-Net algorithm is widely accepted as a representative deep learning model. U-Net was originally developed for the segmentation of biomedical images based on the fully convolutional neural network [19]. The U-Net algorithm can perform excellent pixel-wise semantic segmentation tasks to analyze imagery [24].

In the present study, we adapted the U-Net model of Ronneberger et al. (2015) [25] to use fewer filters, which helps prevent overfitting of the model [26]. We used the gray version of an RGB image as the input and adapted the model accordingly (Figure 3). The U-Net model is composed of an encoding path and a decoding path, which create a U-shaped architecture. The encoding path learns the context of the training images and then the decoding path performs localization, which assigns a class label to each pixel. Specifically, a skip connection joins the encoding path and decoding path to transform local information into global information [24].

### 2.3.4. Random Forest Classification Algorithm

Random forest (RF) is a machine learning algorithm used for classification or regression tasks based on decision trees [27]. In the context of classification, the most frequently voted class from all decision trees is selected as the final prediction Figure 4 [28].

The RF algorithm begins the training process by preparing data using a reference dataset containing features (input images) and corresponding class labels (label masks). Then the algorithm creates a collection of decision trees (ensemble). To build an individual decision tree, the algorithm randomly selects a subset of the training data (with replacement) through the bagging process. Each decision tree in the ensemble is grown using the selected features and the bootstrap sample. The trees are built through recursive partitioning of the data based on the selected features, aiming to create leaves (target classes). When making a prediction for an input data point, each decision tree in the ensemble independently predicts the class label. As noted above, the class predicted through majority voting (trees) is selected as the final prediction. Thus, predictions from all individual trees are

combined to produce a final ensemble prediction. This process helps to reduce overfitting and improve the model's generalizability [29,30].

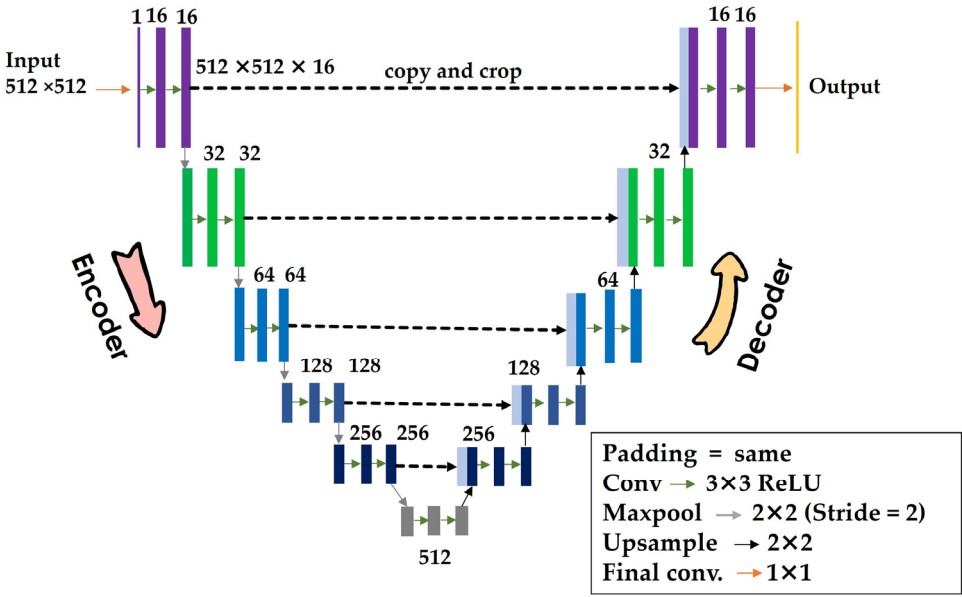

**Figure 3.** U-Net classification algorithm, adapted from [25].

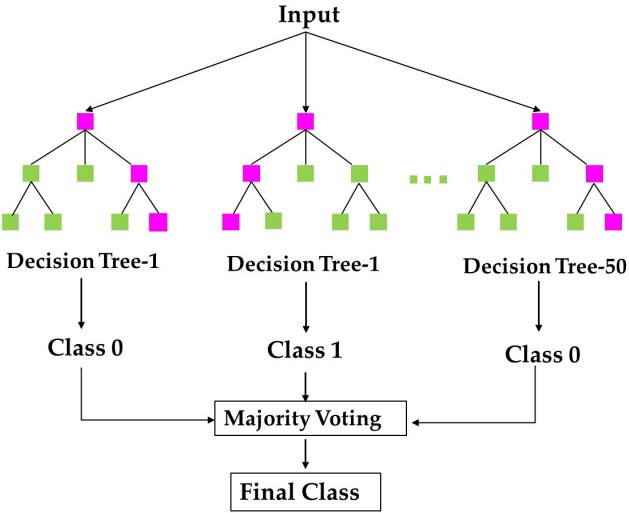

**Figure 4.** Random forest classification algorithm.

2.3.5. Training of the U-Net and RF Models

We used an input image 512 × 512 pixels in size to avoid overloading the computer memory, as mentioned in Section 2.3.2. Moreover, as grayscale images eliminate the influence of color information [31], we converted the RGB UAV orthophotos of both sub-compartments taken in different seasons, into grayscale versions to reduce the impact of color variations on the accuracy of the trained machine learning models. Then the grayscale features were analyzed using the semantic segmentation-based U-Net and RF models. To investigate the importance of data balancing for the training of machine classification models, both imbalanced (containing unequal pixel numbers between the conifer and broadleaf classes) and balanced (containing equal pixel numbers of the conifer and broadleaf classes) datasets were used to train models. We prepared the imbalanced dataset using the UAV dataset of Sub-compartment 42B while the balanced dataset was prepared using the UAV dataset of Sub-compartment 16AB. As broadleaf species are

dominant in Sub-compartment 42B, pixel numbers are greater for the broadleaf class than the conifer class in the imbalanced dataset. For Sub-compartment 16AB, we intentionally selected UAV images that provide balanced pixel numbers for the conifer and broadleaf classes. For all models, 90% of images were used for model development and 10% were retained for validation. To assess how well a machine learning model performs on unseen data, the training dataset was split into a training set (80%) and a testing set (20%). The general workflow of the complete procedure is shown in Figure 5.

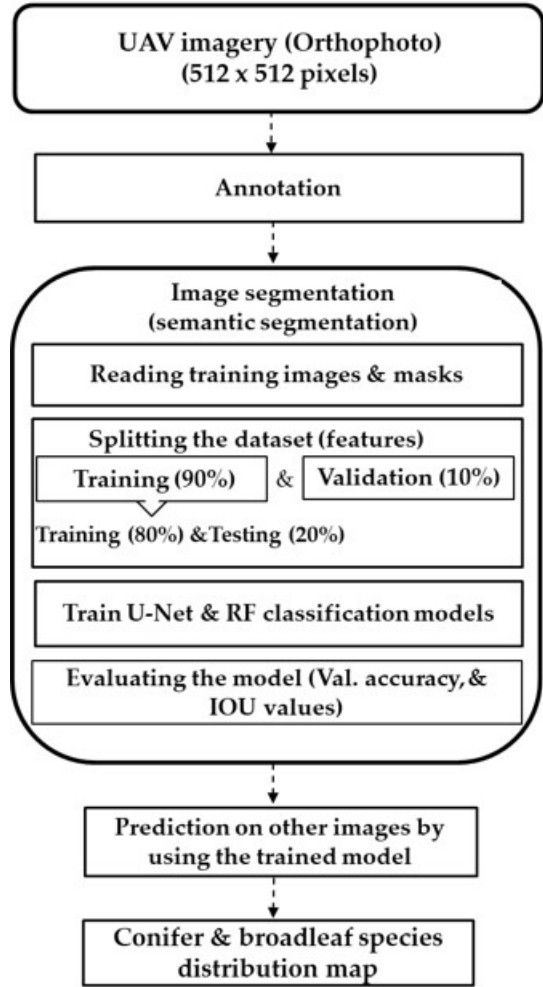

**Figure 5.** General workflow for mapping the distribution of conifer and broadleaf species canopies in uneven-aged forests by using UAV imagery (orthophoto) and machine learning models.

Before training the U-Net model, we set up the initial weights, and activation functions in the layers both in encoding path and decoding path of the model to facilitate effective training. "HeNormal", a kernel_initializer, was used to initialize the weight of network's layers. This kernel_intializer can deal with the problem of vanishing gradients when training deep neural networks. For the activation function in the convolutional layers, we used the Rectified Linear Unit (ReLU), which introduces non-linearity into the feature extraction process, allowing the model to learn complex patterns and representations from the input data [32]. For the output layer, the softmax activation function was chosen for its ability to normalize the output values into a probability distribution, making it suitable for tasks where the model needs to assign each pixel to one of multiple classes [33].

The training of the U-Net models was implemented using the Keras package, with TensorFlow as the backend for Keras [34]. For training of the U-net models, we used 360 images covering the whole area of Sub-compartment 42B. For Sub-compartment 16AB, we used 360 of 1600 available images, in accordance with the number of images used for

Sub-compartment 42B. The total number of pixels representing the conifer and broadleaf classes of both sub-compartments used in the training of U-Net models are presented in Table 1.

**Table 1.** The total number of pixels for conifer and broadleaf classes of Sub-compartments 42B and 16AB to train the machine learning models.

| Machine Learning Models | Sub-Compartments | Class | Pixels |
|---|---|---|---|
| U-Net | 42B | Conifer | 11,601,650 |
| | | Broadleaf | 54,573,657 |
| | 16AB | Conifer | 33,209,707 |
| | | Broadleaf | 33,204,081 |
| RF | 42B | Conifer | 2,941,142 |
| | | Broadleaf | 8,665,505 |
| | 16AB | Conifer | 5,155,835 |
| | | Broadleaf | 5,103,208 |

During U-Net model training, we used the Adam optimizer. The loss function was categorical_crossentropy. To assess the importance of the relationship between the number of epochs and the overfitting or underfitting of the model, the model was trained using 50, 700, and 1000 epochs for the Sub-compartment 42B dataset and at 50, 300, and 500 epochs for the Sub-compartment 16AB dataset, as these numbers provided high scores for validation accuracy and intersection over union (IoU) values after several test runs. In both cases, each epoch is composed of 16 batches with verbosity of 1.

For the RF algorithm, models could not be trained using the complete training set due to computing memory restrictions. Therefore, we selected some of the 360 images used to train the U-Net models, specifically 50 images that yielded the highest IoU values for both classes after several test trials. The total pixel numbers of these 50 images used for the conifer and broadleaf classes in both sub-compartments are shown in Table 1.

To train the RF models, we used the scikit-learn random forest classifier and set two parameters: random_state, which is used to set the seed for the random number generator that controls randomness during the training process; and n_estimator, which specifies the total number of trees to be grown in the RF ensemble. Based on previous reports and our trials, we set values of 42 for random_state and 50 for n_estimator.

2.3.6. Evaluation Metrics for the Models

We evaluated the models using two performance metrics. The first metric was validation accuracy, which measures how accurately the model's predictions match the actual known outcomes for a set of data that the model was not exposed to during training. Meanwhile, the second metric was the IoU value of the object class, which is a valuation metric that measures the overlap between a predicted region (such as a segmented area) and a ground truth region (the actual target region) based on the ratio of the intersection area to the union area of the two regions. The validation accuracy and IoU are defined by the following equations:

$$\text{Validation accuracy} = (\text{Number of Correct Predictions})/(\text{Total Number of Predictions}), \tag{1}$$

$$\text{IoU} = (\text{Area of Intersection})/(\text{Area of Union}) \tag{2}$$

For deep learning models, overfitting can be reduced if the differences between the training and validation accuracies and losses are as small as possible. Therefore, when evaluating the U-Net model, we considered the training accuracy, training loss, and validation loss in addition to the validation accuracy and IoU values described above. Training accuracy is a metric that measures the performance of the model on the training data during the training process and is generally expressed as a percentage representing the ratio of correctly predicted labels to the actual labels of the training data. Meanwhile, validation accuracy is a metric that assesses how well the model generalizes to unseen data

and is calculated in the same manner as training accuracy but with a separate validation dataset that was not used during the training process. The training and validation losses are metrics that quantify the error between the model's predictions and actual data.

In addition, we constructed confusion matrices on the validation dataset as heat maps to visualize the performance of machine learning models in predicting the true classes. Moreover, we also calculated precision and recall, which can be obtained for the error matrix. Precision is the ratio of positive samples that are predicted correctly to the total predicted positives, while recall is the ratio of positive samples that are predicted correctly to the total positive samples. Precision and recall were calculated as follows:

$$\text{Precision} = TP/(TP + FP), \tag{3}$$

$$\text{Recall} = TP/(TP + FN) \tag{4}$$

where

TP (True Positives): Tree crowns are correctly predicted as positive.
FP (False Positives): Tree crowns are falsely predicted as positive.
FN (False Negatives): Tree crowns are falsely predicted as negative when, they are positive.

To obtain insights into the intensity characteristics of areas where the models made errors in the test images, we assessed the mean intensity and standard deviation (SD) of misclassified areas; the former represents the mean intensity within regions where the predicted segmentation differs from the ground truth, while the latter measures the extent to which the intensity values within error-prone regions deviate or vary from the mean intensities within those regions. In grayscale images, pixel intensity typically ranges from 0 (black) to 1 (white), with intermediate values representing shades of gray.

2.3.7. Algorithm

Both the U–Net and RF models were coded in the Spyder environment (version 5.1.5) using the Python language. The training time of the U–Net models for each sub-compartment's UAV dataset was 12 h for 500 epochs and 1 h for 50 epochs, while the training time of the RF model was 1 h using a 2ML54VSB laptop with an AMD Ryzen 7 4800H processor and 16.0 GB RAM.

**3. Results**

*3.1. Discrimination between Coniferous and Broadleaf Canopy Cover Using an Imbalanced UAV Dataset with Machine Learning (U–Net and RF) Models*

As described in Section 2.3.5, after we trained the U–Net model with 50, 700, and 1000 epochs for Sub-compartment 42B, the performance of the model was represented by validation accuracies of 0.88, 0.93, and 0.93 with mean IoU of 0.57, 0.80, and 0.79, respectively. Among the three models, the validation accuracy scores and IoU values of the model at 50 epochs were the lowest. Meanwhile, the models with 700 and 1000 epochs performed very similarly in terms of validation accuracy and IoU values for the broadleaf class. However, the mean IoU value and IoU value for the conifer class (0.80 and 0.60) were higher for the model with 700 epochs than with 1000 (0.79 and 0.54, respectively) (Table 2).

**Table 2.** Validation accuracies and IoU values of discriminating conifer and broadleaf species canopies cover in Sub-compartment 42B by using UAV imagery and machine learning models.

| **Machine Learning Models** | | **Validation Accuracy** | **IoU (Mean)** | **IoU (Conifer)** | **IoU (Broadleaf)** |
|---|---|---|---|---|---|
| U–Net | 50 epochs | 0.88 | 0.57 | 0.00 | 0.78 |
| | 700 epochs | 0.93 | 0.80 | 0.60 | 0.86 |
| | 1000 epochs | 0.93 | 0.79 | 0.54 | 0.86 |
| RF | | 0.69 | 0.33 | 0.04 | 0.68 |

In addition, although the differences between training and validation accuracies were similar for the two models, the difference between training and validation losses was smaller for the model with 700 epochs (0.063 and 0.395) than the model with 1000 epochs (0.063 and 0.492) (Figure 6). Meanwhile, the difference between the training and validation accuracies and the difference between the training and validation losses of the model with 50 epochs was the smallest among the three models; however, this model faced an underfitting problem because the validation accuracy increased while the validation loss decreased with increasing number of epochs. Thus, using the aforementioned evaluation metric scores, we selected the model with 700 epochs as the most accurate model in this study. Model performance at prediction of images with different numbers of epochs is visualized in Figure 7c–e.

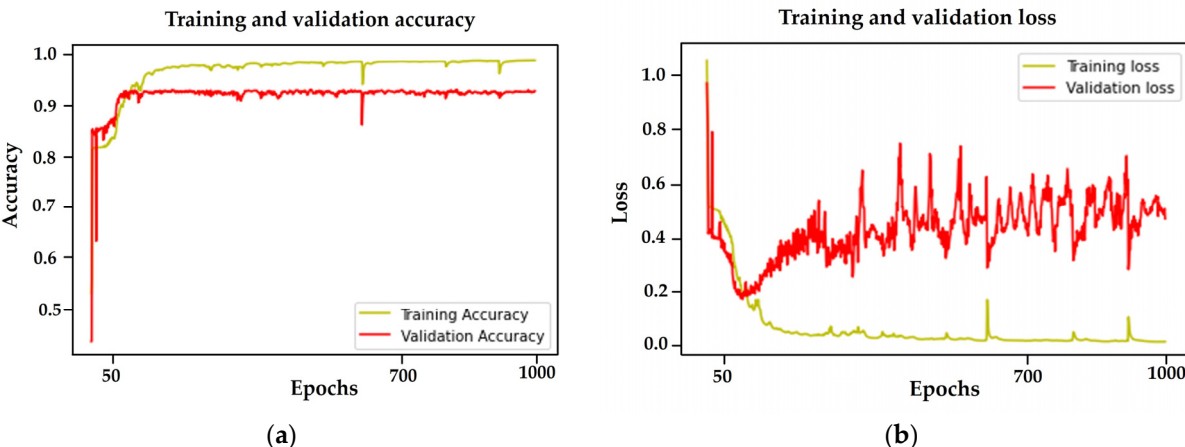

**Figure 6.** (**a**) Training and validation accuracies of U-Net model at different epochs. (**b**) Training and validation losses of U-Net model at different epochs. (Sub-compartment 42B).

After training of the RF classification model, the performance of the model was moderate in terms of validation accuracy (0.69). The RF classifier performed well at discriminating the broadleaf class based on IoU value (0.68). However, the IoU value for classification of the conifer class (0.04) indicated very poor performance. Compared to the U-Net model (with 700 epochs), performance of the RF model was poorer based on the evaluation metrics (Table 2). This result was also confirmed through visual comparison as shown in Figure 7.

### 3.2. Discrimination between Coniferous and Broadleaf Canopy Cover Using a Balanced UAV Dataset with Machine Learning (U-Net and RF) Models

Among the U-Net models employing 50, 300 and 500 epochs for Sub-compartment 16AB, the model with 300 epochs performed best, with validation accuracy of 0.83 and IoU values of 0.70 for the mean, 0.70 for the conifer class and 0.69 for the broadleaf class (Table 3). The performance of the U-Net model with 500 epochs was moderate, while that of the model with 50 epochs was poorest in terms of validation accuracy and IoU values (Table 3). The differences between training and validation accuracies and losses were smaller for the model with 300 epochs than for the model with 500 epochs, while the model at 50 encountered underfitting (Figure 8) as like in Section 3.1. Therefore, in terms of the highest scores of evaluation metrics, the model with 300 epochs was accepted as the optimal model. The prediction results of test images using the U-Net models with different epochs is shown in Figure 9.

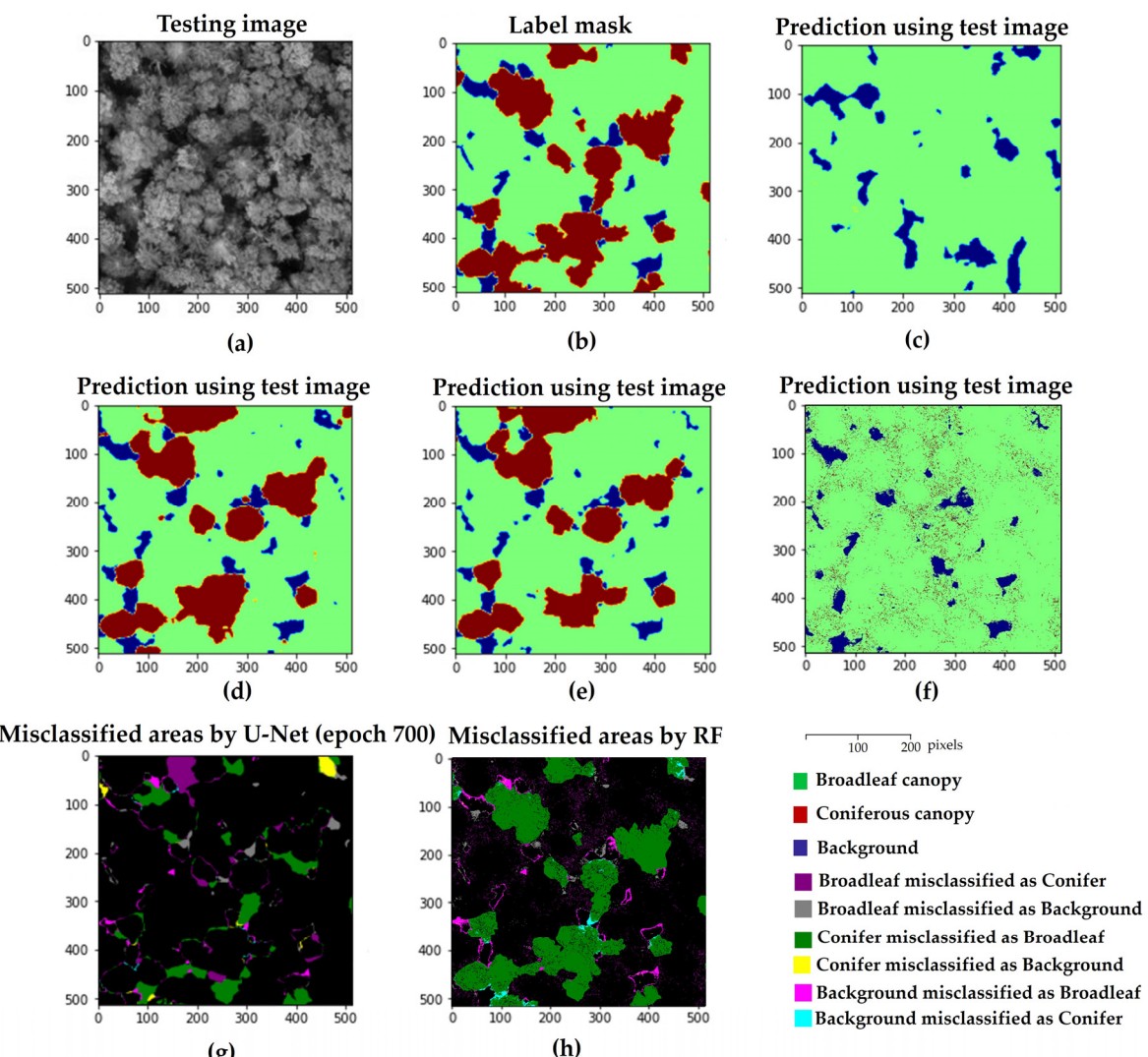

**Figure 7.** Visualization of the predicted results of canopy cover mapping by the machine learning models (Sub-compartment 42B). (**a**) Testing grayscale image; (**b**) label mask; (**c**) prediction using the test image by the U-Net model with 50 epochs; (**d**) prediction using the test image by the U-Net model with 700 epochs; (**e**) prediction using the test image by the U-Net model with 1000 epochs; (**f**) prediction using the test image by the RF model; (**g**) misclassified areas predicted by the U-Net model (with 700 epochs); and (**h**) misclassified areas predicted by the RF model.

**Table 3.** Validation accuracies and IoU values of discriminating conifer and broadleaf species canopies cover in Sub-compartment 16AB by using UAV imagery and U-Net models at different epochs.

| Machine Learning Models | | Validation Accuracy | IoU (Mean) | IoU (Conifer) | IoU (Broadleaf) |
|---|---|---|---|---|---|
| U-Net | 50 epochs | 0.75 | 0.61 | 0.62 | 0.55 |
| | 300 epochs | 0.83 | 0.70 | 0.70 | 0.69 |
| | 500 epochs | 0.79 | 0.66 | 0.67 | 0.61 |
| RF | | 0.59 | 0.43 | 0.40 | 0.42 |

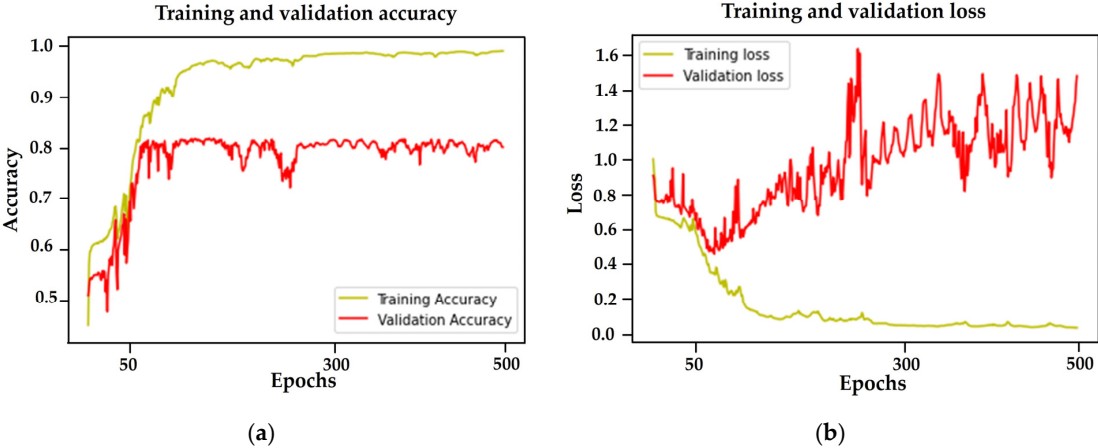

(**a**)

(**b**)

**Figure 8.** (**a**) Training and validation accuracies of U-Net model at different epochs. (**b**) Training and validation losses of U-Net model at different epochs. (Sub-compartment 16AB).

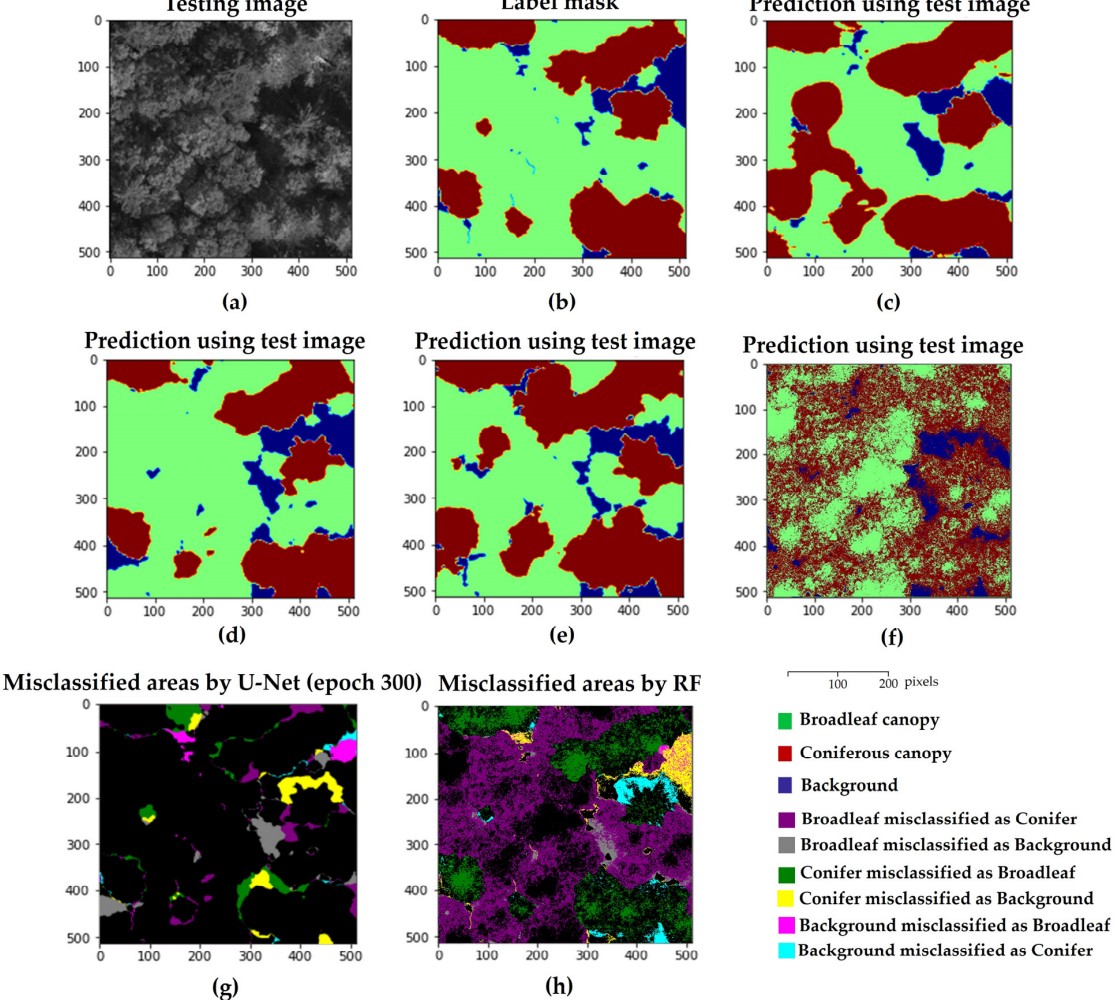

**Figure 9.** Visualization of the predicted results of canopy cover mapping by the machine learning models (Sub-compartment 16AB). (**a**) Testing grayscale image; (**b**) label mask; (**c**) prediction using test image by the U-Net model with 50 epochs; (**d**) prediction using test image by the U-Net model with 300 epochs; (**e**) prediction using test image by the U-Net model with 500 epochs; (**f**) prediction using the test image by the RF model; (**g**) misclassified areas predicted by the U-Net model (with 300 epochs); and (**h**) misclassified areas predicted by the RF model.

When we trained the RF classification model using the balanced dataset, the results obtained were a validation accuracy of 0.59 and mean IoU of 0.43, with values of 0.40 for the conifer class and 0.42 for the broadleaf class. The performance of the model at classification of the conifer class was very similar to performance for the broadleaf class. Compared to the RF model using the imbalanced dataset described in Section 3.1, the mean IoU and IoU value for the conifer class showed improvements, although the validation accuracy and the IoU value for the broadleaf class were lower (Table 3).

Nevertheless, relative to the U-Net model (with 300 epochs) using the same balanced dataset, the RF model performed poorly in terms of all evaluation metrics (Table 3) as well as visual interpretation (Figure 9).

*3.3. Mapping the Distribution of Coniferous and Broadleaf Canopy Cover in Uneven-Aged Forests through Integration of UAV Imagery with Machine Learning Technology*

According to the confusion matrix heat map (Figure 10a), the U-Net model (with 700 epochs) trained with the Sub-compartment 42B (imbalanced) UAV dataset could predict both the conifer class (60.74%) and broadleaf class (96.40%) with an average precision of 0.92 and an average recall of 0.85 (Table 4), with higher performance at broadleaf classification. However, the model also created some misclassified areas (mean intensity of 0.04 and SD of 0.01). Overall, 37.32% and 1.94% of the conifer class were incorrectly classified as broadleaf and background classes. On the other hand, 1.62% and 1.98% of the broadleaf canopies were incorrectly detected as conifer canopy and background, respectively. In addition, a few percent of the background class, including some shadow areas, was erroneously assigned to the conifer and broadleaf classes (0.12% and 2.82%, respectively). An example of areas in Sub-compartment 42B that were misclassified by the U-Net model (with 700 epochs) is presented in Figure 7g.

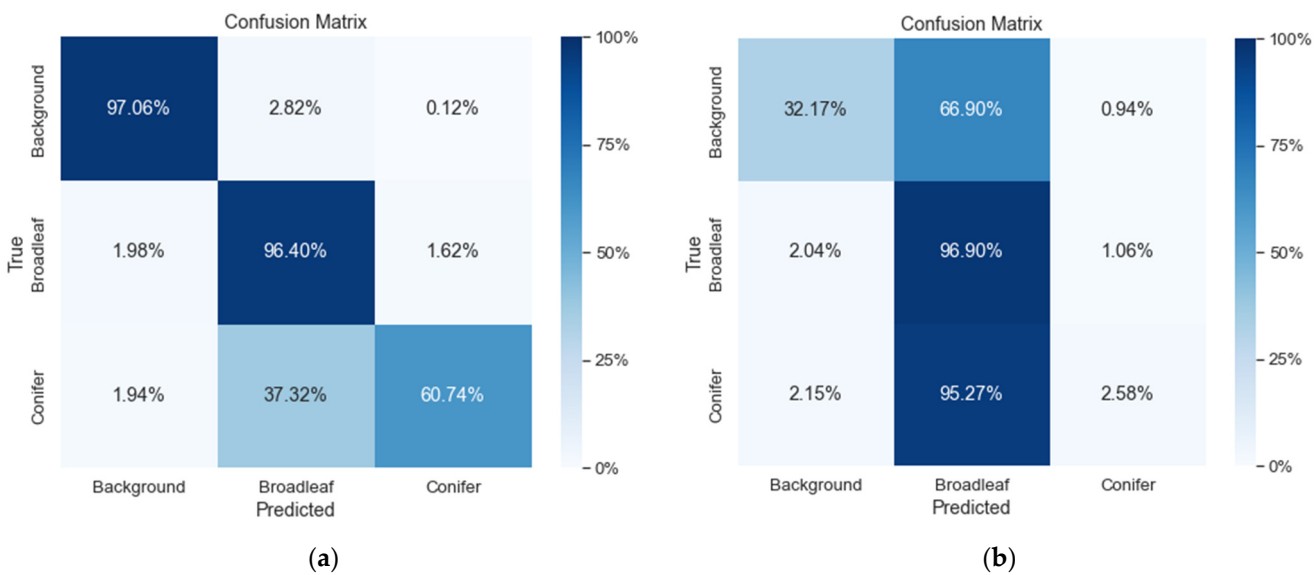

**Figure 10.** Confusion matrix of tree species group segmentation in Sub-compartment 42B through machine learning models; (**a**) by U-Net model (with 700 epochs); and (**b**) by the RF model.

**Table 4.** Average precision and recall values of the selected optimal U-Net models and RF models to discriminate conifer and broadleaf species canopies cover in both Sub-compartments 42B and 16AB.

| Machine Learning Models | Sub-Compartments 42B | | Sub-Compartments 16AB | |
|---|---|---|---|---|
| | Precision | Recall | Precision | Recall |
| U-Net | 0.92 | 0.85 | 0.83 | 0.83 |
| RF | 0.60 | 0.44 | 0.61 | 0.60 |

Meanwhile, the confusion matrix heat map (Figure 10b) showed that the RF model performed very well at classifying the true broadleaf class (96.90%), with only a small proportion of broadleaf areas misclassified (1.06% as conifer and 2.04% as background). However, 95.27% and 2.15% of the conifer class was misclassified into the broadleaf and background classes, while accurate prediction of the conifer class accounted for only 2.58%. Moreover, 66.90% of the background class was misclassified into the broadleaf class. The visual interpretation of areas misclassified created by the RF model is presented in Figure 7h. The mean intensity of misclassification by the RF model was 0.32, with SD of 0.39. Predictions across the entire Sub-compartment 42B by the U-Net (with 700 epochs) and RF models are shown in Figure 11a,b).

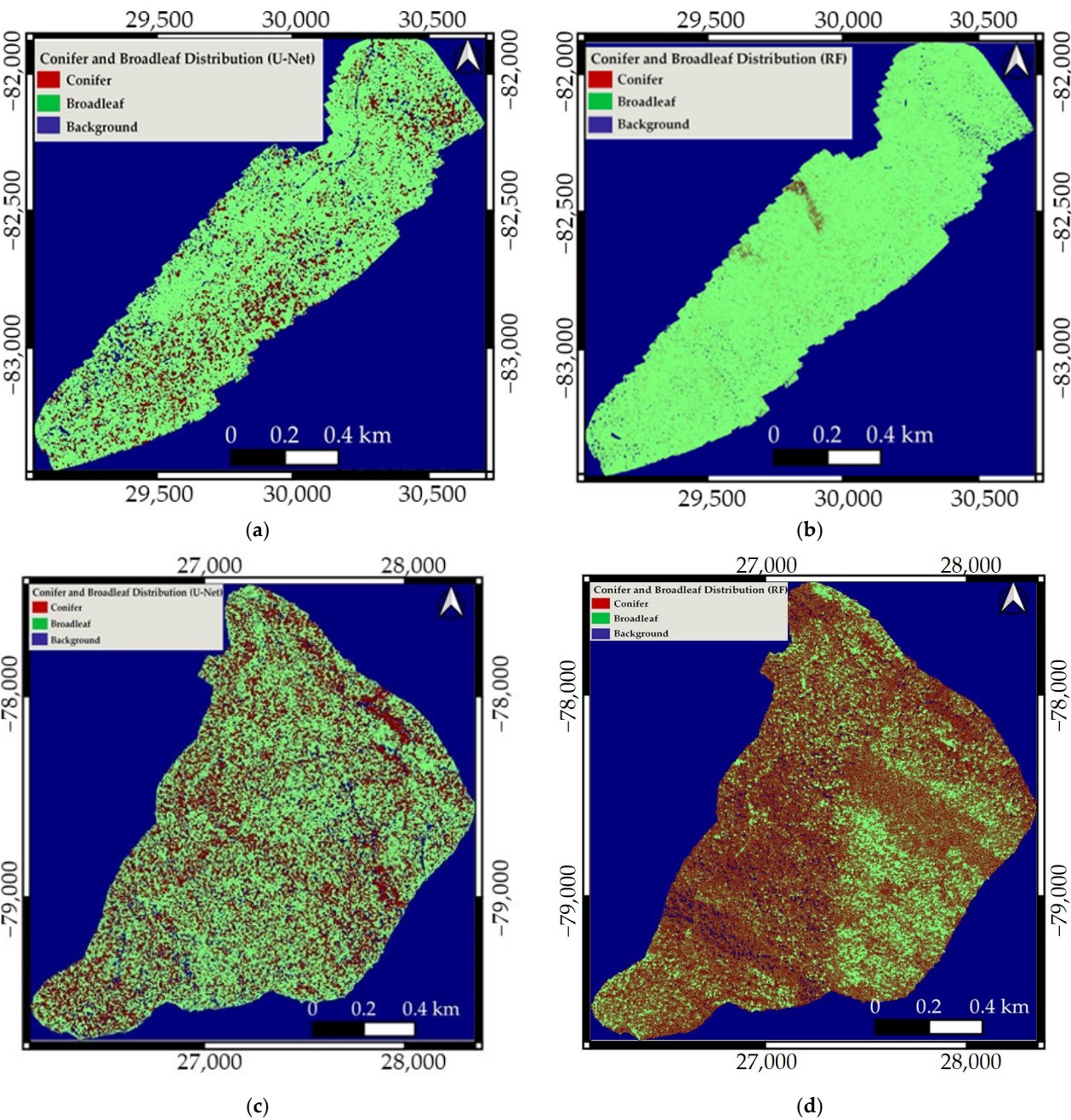

**Figure 11.** Visualization of the predicted Sub-compartments 42B and 16AB. (**a**) Predicted Sub-compartment 42B by the U-Net model (with 700 epochs); (**b**) predicted Sub-compartment 42B by the RF model; (**c**) predicted Sub-compartment 16AB by the U-Net model (with 300 epochs) and (**d**) predicted Sub-compartment 16AB by the RF model.

When the U-Net model (with 300 epochs) was trained using the 16AB (balanced) UAV dataset, the model performed well at predicting both the broadleaf and conifer classes, with 85.34% correct prediction for the broadleaf class and 79.69% for the conifer class as seen in confusion matrix (Figure 12a). Meanwhile, the average precision and recall were both 0.83 (Table 4). However, similar to the results obtained with the imbalanced dataset, the U-Net model (with 300 epochs) had incorrect predictions in some areas, including shadow areas, with a mean intensity of 0.04 and SD of 0.02. The confusion matrix (Figure 12a) indicated that 15.58% and 4.74% of the conifer class were misclassified into the broadleaf and background classes, 10.85% and 3.81% of the broadleaf class were misclassified into the conifer and background classes, and 6.85% and 10.36% of the background class were misclassified into the conifer and broadleaf classes, respectively. Some areas that were misclassified by the U-Net model (with 300 epochs) are shown in Figure 9g.

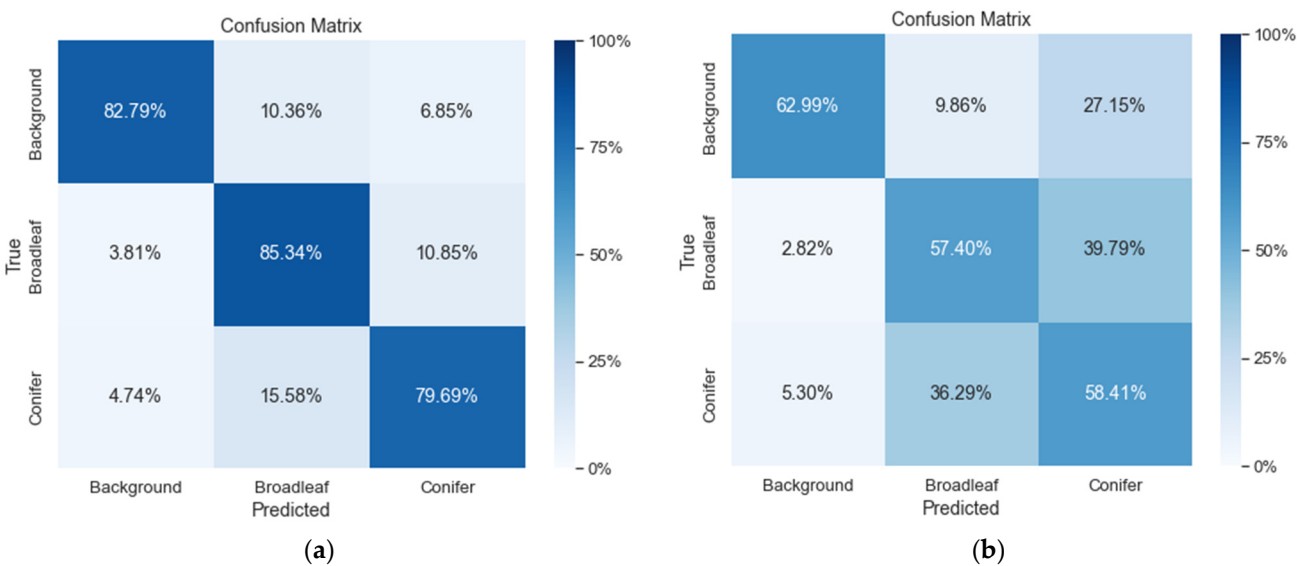

**Figure 12.** Confusion matrix of tree species group segmentation in Sub-compartment 16AB through machine learning models; (**a**) by U-Net model (with 300 epochs); and (**b**) by the RF model.

Using the balanced dataset, the RF model accurately segmented only 57.40% of the broadleaf class and 58.41% of the conifer class (Figure 12b) with mean precision and recall of 0.61 and 0.60, respectively (Table 4). Moreover, 39.79% of broadleaf canopies were misclassified as conifers and 36.29% of conifers were erroneously classified as broadleaf canopies, while 27.15% and 9.86% of the background class were mistakenly classified as conifer and broadleaf canopies. The misclassified areas predicted by the RF model had a mean intensity of 0.28 and SD of 0.37, are shown in Figure 9h. The prediction results across the entire area of Sub-compartment 16AB from both the U-Net and RF models are visualized in Figure 11c,d.

## 4. Discussion

We used the U-Net and RF models to segment the conifer and broadleaf species groups in uneven-aged mixed forests. We found that the U-Net model was more reliable for integration with UAV imagery (both imbalanced and balanced datasets) according to the evaluation metrics summarized in Tables 2 and 3 and visualized in Figures 7 and 9. Previous studies [35–38] have also revealed that deep learning algorithms, including U-Net and other convolutional neural networks, are more applicable than traditional machine learning algorithms such as RF, support vector machine (SVM), and local maxima for image classification tasks because they can learn more diverse and representative features, while traditional machine learning models may struggle to efficiently handle massive datasets. Moreover, these deep learning models can capture complex, nonlinear relationships in the data, which often contain intricate patterns and relationships that may pose challenges

for linear classifiers or decision tree-based models such as RF. The sematic segmentation-based U-Net model can automatically learn hierarchical features of the image that help to capture the context, shape, and relationships of objects in the image, which are crucial for accurate segmentation.

When compared to past studies of tree species mapping in uneven-aged mixed forests, the accuracy scores obtained in the present study for the coniferous and broadleaf species groups using UAV imagery and U-Net models (0.93 in Sub-compartment 42B and 0.83 in Sub-compartment 16AB) were higher than the score obtained by Abdollahnejad and Panagiotidis (2020) [39], who mapped tree species in a mixed conifer–broadleaf forest using unmanned aircraft system multispectral imaging and SVM (0.81). Moreover, our results were also comparable to those of Yang et al. (2021) [40], who estimated the conifer–broadleaf ratio in a mixed forest based on Gaofen-1 (GF-1) normalized difference vegetation index time-series data and the semi-supervised k-means clustering method (0.84).

In the present study, we trained both the U-Net and RF models using the imbalanced UAV dataset of Sub-compartment 42B, and both models provided better performance at segmenting the broadleaf class during classification into conifer and broadleaf classes, as presented in Table 2 and the confusion matrix (Figure 10). This phenomenon was most likely due to the imbalanced dataset, as the number of pixels was greater for the broadleaf class than for the conifer class in this analysis (Table 1). When we used the balanced UAV dataset in Sub-compartment 16AB, the accuracy score of segmenting the conifer class improved, as noted in Table 3 and the confusion matrix (Figure 12). Indeed, machine learning models such as U-Net and RF focus on the majority class and struggle to learn the minority class effectively when the dataset is imbalanced. This narrow focus may result in biased predictions [41–44].

Our findings are in accordance with previous research [45] that has found that a balanced UAV dataset provides superior performance (accuracy of 0.94–0.99) compared to an imbalanced one (accuracy of 0.74–0.78) for detecting healthy fir trees using deep learning models. Moreover, we found that the validation accuracy and IoU value of the broadleaf class decreased when we used the balanced UAV dataset for training of the machine learning models, although these values were high when we used the imbalanced dataset. This difference might be due to an insufficient number of pixels in the training dataset. Thus, we considered a balanced dataset with sufficient pixel number to be important for training machine learning models with the purpose of tree species group segmentation tasks. Future research should consider using balanced datasets with sufficient pixel numbers to improve upon existing research. Moreover, if it is not possible to prepare a perfectly balanced dataset for training machine learning models, the use of an oversampling technique for the minority class to balance the class distribution and a weighted loss function to increase the weight of the minority class during training should be considered. A possible alternative is to use a transfer learning technique, where a CNN model is trained on large or diverse datasets such as ImageNet or GoogLeNet [46]. The pretrained model serves as a starting point for a new task [47] and can help mitigate the challenges of limited labelled data in imbalanced datasets. By using the knowledge gained during pre-training, the model may require less labelled data to adapt to the specific unbalanced task.

In the context of the U-Net algorithm, when the model was trained on the imbalanced dataset with 50 epochs, it failed to distinguish conifer cover, as shown in Figure 7c. This failure may have occurred because the conifer class is underrepresented in the training data compared to other classes, and thus the model may have struggled to detect that class. As training continues and more epochs are allowed, the model can gradually adapt and learn to recognize the minority class more effectively. Pyo et al. (2022) [24] suggested that the number of epochs should be considered when selecting the optimal U-Net model based on semantic segmentation. In fact, the ModelCheckpoint callback function provided in the Keras library, which is an open-source deep learning library written in Python, was designed to provide an option to save the best model based on a monitored metric at a specific epoch number. However, this function works on a single monitored metric (i.e.,

either validation loss or validation accuracy). Therefore, our further experiment will focus on improving this technique to use composite metrics.

The U-Net models selected as optimal in this study produced some misclassifications, as noted in Section 3.3. Given the mean intensity values of 0.04 for both imbalanced and balanced datasets with SDs of 0.01 and 0.02, respectively, the U-Net model may be weak at prediction in dark regions with low variation in pixel intensity within the misclassified areas. Meanwhile, the RF model provided a mean intensity of 0.32 with SD of 0.39 for the imbalanced dataset and mean intensity of 0.28 with SD of 0.37 in the balanced dataset for misclassified area. These values suggest that, on average, the misclassified areas have a range of gray tones with moderate levels of variability around the mean. This interpretation was supported by the visual comparison presented in Figures 7 and 9, as the U-Net models sometimes incorrectly classified shadow areas as tree canopies, while the RF models were able to predict such areas correctly.

The misclassification between conifer and broadleaf canopies may be related to the complexity of uneven-aged mixed forests. Uneven-aged mixed forests with intermingled forest conditions and high canopy closure create a challenge for the models when matching a label to a specific pixel in the image. Misclassified areas can be reduced by adding enhanced data for these problematic areas (i.e., additional data under a variety of lighting conditions, texture information) to the grayscale features. Moreover, regarding the misclassification of coniferous canopies as broadleaf canopies, Onishi et al. (2022) [48] noted that the canopies of coniferous trees are not readily distinguished from broadleaf tree canopies in UAV images taken more than 100 m above ground level. In our study, the UAV imagery was taken from 130 m above ground level, providing a possible reason for misclassification between coniferous canopies and broadleaf canopies.

Nevertheless, in terms of accuracy scores, our results for segmenting coniferous and broadleaf species groups using grayscale UAV imagery and U-Net models (validation accuracies: 0.93 in Sub-compartment 42B, 0.83 in Sub-compartment 16AB) were better than the results obtained by Schiefer et al. (2020) [18] for the mapping of tree species in heterogenous temperate forests using high-resolution UAV-based RGB imagery and U-Net models (F1 score, 0.73). Moreover, using single-channel grayscale images as input to train the U-Net model not only eliminated the influence of color information, but also reduced the computational complexity of the models and simplified the architecture of the U-Net model, making training and inference faster and requiring less memory. We were able to handle large UAV datasets with the U-Net model using a personal laptop with 16.0 GB of RAM. However, it is important to note that the choice of single or multi-channel inputs depends on the specific requirements of the task. In applications where color information is critical, the use of multi-channel inputs (e.g., RGB) may be more appropriate.

Zhong et al. (2022) [49] achieved higher accuracy from fused light detection and ranging (LiDAR) and UAV-based hyperspectral data (accuracy = 89.20%) than from either LiDAR (accuracy = 76.42%) or hyperspectral data (accuracy = 86.08%) alone in the context of individual tree species identification in natural mixed conifer and broadleaf forests. In addition, Ye et al. (2021) [29] highlighted the applicability of spectral features, textural features, and tree height information for vegetation classification in complex mixed forests. Adding such information to the information used in this study could improve the accuracy of discrimination between coniferous and broadleaf canopies.

## 5. Conclusions

This study focused on the applicability of the integration of UAV imagery with the semantic segmentation-based U-Net and RF classification models to the mapping of conifer and broadleaf canopy cover in uneven-aged mixed forests. The grayscale features of RGB images and corresponding label layers were analyzed for the classification tasks. We trained the models using the UAV imagery datasets of Sub-compartment 42B (imbalanced pixel numbers between the conifer and broadleaf classes) and Sub-compartment 16AB (balanced pixel numbers between the conifer and broadleaf classes) of UTHF.

This case study revealed that the integration UAV imagery with the U-Net model allows for mapping of the distribution of conifer and broadleaf canopy cover in complex uneven-aged mixed forests with reliable accuracy. These findings establish a practical methodology for obtaining precise and up-to-date information about the dominant tree species in uneven-aged mixed forests. In addition, this study demonstrates that using a balanced dataset can reduce misclassification of the conifer class, which was poorly represented in the imbalanced dataset. Moreover, our results reveal that the U-Net model made errors in dark regions of the grayscale version of the RGB UAV images. These findings provide useful information for the forest managers to improve tree species groups segmentation tasks in uneven-aged mixed forests.

As we used only grayscale information from RGB UAV images, our future research will consider data fusion after adding other information such as texture and tree height to improve the performance of the trained models for classifying tree species groups in complex uneven-aged mixed forests. Moreover, as transfer learning techniques can help the models learn hierarchical features and patterns related to canopy structure, further research will use pretrained models on large datasets to reduce the problem of misclassification between conifer and broadleaf canopies in uneven-mixed forests.

**Author Contributions:** Conceptualization, methodology, formal analysis, and writing—original draft preparation, N.M.H.; resources(i.e., UAV datasets and the necessary resources for data analysis), supervision, and writing—review and editing, T.O.; review and editing, S.T. and T.H. All authors have read and agreed to the published version of the manuscript.

**Funding:** This research received no external funding.

**Data Availability Statement:** The UAV datasets and codes in the present study will be available on request to the corresponding author's email with appropriate justification.

**Acknowledgments:** The authors would like to acknowledge the technical staff of the University of Tokyo Hokkaido Forest (UTHF); Koichi Takahashi, Satoshi Chiino, Yuji Nakagawa, Takashi Inoue, Masaki Tokuni, and Shinya Inukai, for collecting the UAV imagery of the Sub-compartment 42B and 16AB as well as helping in our visual labelling of the orthophotos.

**Conflicts of Interest:** The authors declare no conflict of interest.

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
