# Peer review of "Integration of Unmanned Aerial Vehicle Imagery and Machine Learning Technology to Map the Distribution of Conifer and Broadleaf Canopy Cover in Uneven-Aged Mixed Forests"

_drones, doi:10.3390/drones7120705_

Round 1

Reviewer 1 Report

Comments and Suggestions for Authors

The manuscript proposed a machine-learning-based method for semantic segmentation of conifer and broadleaf canopies in mixed forests using UAV imagery. The paper is properly organized and the results are well presented. Two chosen machine learning methods achieved good results in two UAV imagery datasets. It can be recommended for publication after major revisions. Some details of this manuscript need more interpretation, as following.

Line 146. Please provide more detail about the manual densely-labelling workload required by the RF-based segmentation tool.

Line 194. Please include addtional description about the initialization strategy of layers in U-Net model. If grayscale images are taken as inputs for U-Net, the first layer of U-Net needs to be adapted to single-channel input, while pretrained weights for U-Net with RGB inputs cannot be utilized. To our best knowledge, utilizing pretrained weights can greatly improve the segmentation performance for popular neural network models including U-Net. This choice of using single-channel input also needs more discussion.

Line 296. During training iterations, rather than selecting from just 3 validation timing, a criterion should be set for saving the better-performing model via intermittent validation.

Line 369. Precision, Recall, and Overall Accuracy in the final test for both RF and U-Net model is suggested to be included in a table, in addtion to the confusion matrix already presented, so that readers can get a clear picture about segmentation performance.

Comments on the Quality of English Language

Minor editing of English language required

Reviewer 2 Report

Comments and Suggestions for Authors

The Manuscript titled “Integration of Unmanned Aerial Vehicle Imagery and Machine Learning Technology to Map the Distribution of Conifer and Broadleaf Canopy Cover in Uneven-Aged Mixed Forests” builds upon the literature base to introduce semantic segmentation for describing conifer and broadleaf forest canopies, focusing on U-Net and Random Forest classifiers. The authors found U-NET to be the most reliable, both on an imbalanced and balanced dataset, though random forest showed some utility in the balanced dataset. Their research did validate previous findings in other papers, showing better results from a balanced dataset than imbalanced. Overall, I believe the manuscript to be well written and worthy of publication after the correction of some minor revisions. My specific comments are below:

Introduction:

Lines 39-40- can stay same paragraph; no need to make a small paragraph tangent before introducing remote sensing.

Line 49- awkward phrasing- maybe change to “high cost of data acquisition”

Lines 74-81: phrasing makes it sound like there is a case study- which in some sense there is but perhaps rephrasing some statements here would help. Otherwise nice job framing the study!

Materials and Methods

Line 109- Is the Zenmuse also RGB? Might be beneficial for readers to know.

Line117- interesting! This is a separate statement- what are the legal limits of flight in Japan for a UAS? 

Line 124- perhaps showing GCP locations in Figure 1 would help. Where were the GCPS? How were they disperse? Why did you choose 5, not 10-15?

2.3.3 U-Net- excellent description.

Line 173- Excellent description, but a figure here might also help readers.

Lines 242-244- what about users and producers accuracy? Were these considered?

Section 4- Discussion

Much of the discussion is too lengthy. Some if it is already discussed in the results, and then reiterated in the context of other literature in the discussion. I recommend thinning the discussion down a little bit for the reader- what are the most important takeaways from your perspective as authors, and limit it to that.

Round 2

Reviewer 1 Report

Comments and Suggestions for Authors

Accepted